# VisualLens: Personalization through Task-Agnostic Visual History

**Wang Bill Zhu**[◇♥†*]     **Deqing Fu**[◇♥†]     **Kai Sun**[◇]     **Yi Lu**[◇]     **Zhaojiang Lin**[◇]
**Seungwhan Moon**[◇]     **Kanika Narang**[◇]     **Mustafa Canim**[◇]
**Yue Liu**[◇]     **Anuj Kumar**[◇]     **Xin Luna Dong**[◇]

[◇]Meta     [♥]USC

## Abstract

Existing recommendation systems either rely on user interaction logs, such as online shopping history for shopping recommendations, or focus on text signals. However, item-based histories are not always accessible, and are not generalizable for *multimodal recommendation*. We hypothesize that a user's visual history — comprising images from daily life — can offer rich, task-agnostic insights into their interests and preferences, and thus be leveraged for effective personalization. To this end, we propose VISUALLENS, a novel framework that leverages multimodal large language models (MLLMs) to enable personalization using task-agnostic visual history. VISUALLENS extracts, filters, and refines a spectrum user profile from the visual history to support personalized recommendation. We created two new benchmarks, **Google Review-V** and **Yelp-V**, with task-agnostic visual histories, and show that VISUALLENS improves over state-of-the-art item-based multimodal recommendations by 5-10% on Hit@3, and outperforms GPT-4o by 2-5%. Further analysis shows that VISUALLENS is robust across varying history lengths and excels at adapting to both longer histories and unseen content categories.

## 1 Introduction

Imagine a personal assistant, similar to Vannevar Bush's MEMEX [4], observing what you do in your daily life. With her keen insight, she can make informed guesses about what you may enjoy or find intriguing. When you ask for recommendations on anything from restaurants and activities to movies, books, and products, based on her in-depth understanding of you she will provide suggestions tailored specifically to your tastes.

While the concept is intuitive, a truly comprehensive personal assistant capable of making recommendations across all aspects of life has yet to be realized. Most existing multimodal recommendation systems remain domain-specific and rely heavily on item-based interaction histories [41, 34]. For example, an e-commerce platform may suggest products based on past purchases but ignore dining habits or interests outside shopping.

This work explores *how can we leverage such a user's visual record* to better understand individual preferences and enable more general, personalized recommendations. Achieving task-agnostic recommendations from visual history poses several challenges. First, visual histories are often *diverse and noisy*, containing images unrelated to any specific recommendation task, entities that fail to accurately represent user preference, or non-informative elements (*e.g.*, background objects like

---

[*]Correspondence: `wangzhu@usc.edu`
[†]Work done at Meta.

39th Conference on Neural Information Processing Systems (NeurIPS 2025).

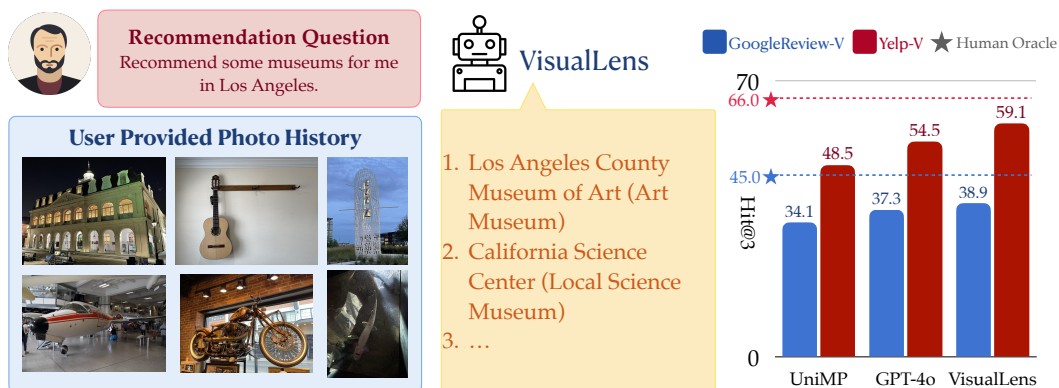

Figure 1: VISUALLENS leverages a user's task-agnostic visual history to provide personalized recommendations. Our method outperforms GPT-4o by 1.6%~4.6% on Hit@3.

trash cans). This creates a trade-off between preserving rich visual content and extracting clean, interpretable representations of user interests. Second, most current MLLMs can only process a *limited number* of images, requiring selective retrieval of relevant user profile information for each query. Third, existing benchmarks are *inadequate for evaluating* task-agnostic visual recommendation systems, highlighting the need for new, purpose-built evaluation datasets.

We propose a novel framework VISUALLENS, as a first step towards harnessing a user's visual history for MLLM recommendation. VISUALLENS begins by extracting an offline spectrum user profile, compressing each image in the visual history into a triplet: (*raw image, caption, aspect words*). This representation spans a spectrum from rich but noisy content (*raw image*) to concise but clean semantic cues (*aspect words*). To improve the quality of aspect words, we employ an iterative refinement process that progressively enhances their alignment with user interests. Next, to efficiently incorporate multiple images during runtime recommendation, VISUALLENS retrieves the most *relevant* segments of the user profile based on the query context. These selected images are organized into a $d \times d$ visual grid, accompanied by their corresponding captions and aspect words, enabling the model to jointly process and reason over a compact yet informative representation of the user's visual history. Finally, we train the system to perform both aspect refinement and recommendation question-answering (QA) in a *unified* model. This design not only reduces parameter overhead but also strengthens the model's ability to interpret and utilize visual history for accurate, personalized recommendations.

To facilitate the evaluation, we created two new benchmarks, *Google Review-V* and *Yelp-V*, providing a foundation for personalization assessments. We leverage user-taken photos to address the challenge of history availability. Unlike extensive MEMEX video logs, these photos require less storage, often available in reviews and social media posts, and provide more insights into a user's interests and preferences. Each benchmark includes a standard test set targeting generalization to *new users*, along with two additional test sets for transferring to *longer histories* and *unseen categories*.

Our experimental study shows promising recommendation quality of VISUALLENS. It achieved 82-91% Hit@10 on the Google Review-V and Yelp-V benchmarks, outperforming state-of-the-art (UniMP [46]) by ~10%. Even comparing with GPT-4o, our 8B model improves Hit@3 by 1.6% and 4.6% respectively on the two benchmarks. Further analysis reveals that VISUALLENS excels in adapting to longer histories and unseen categories, while maintaining robustness with shorter histories.

## 2 Related Works

**Recommendation system with large language models.** Large Language Models (LLMs) have demonstrated strong potential in recommendation systems with their advanced language processing capabilities [41]. On item-based recommendation, studies such as LLM4RS [6], LLMRank [19], CLLM4Rec [62], P5 [12], and Sanner et al. [38] explored various LLM prompt and bootstrapping strategies, showing competitive performance, especially in cold-start scenarios. Generative recommendation with open-domain prompts is explored by GenRec [21], though fine-tuning remains crucial. Fine-tuning approaches include personalized aspect extraction [26], and multitasking on

Table 1: Unlike representative related works, VISUALLENS is a novel framework leveraging multimodal LLM for recommendation with task-agnostic visual content. CTR: click-through rate.

| | Multimodal | New Eval | User History | User Profile | Objective |
|---|---|---|---|---|---|
| LLM4RS [6] | ✗ | ✗ | Textual item features | Text sequence | Items |
| ReLLa [27] | ✗ | ✗ | Textual item features | Top-k behaviors | CTR |
| ONCE [29] | ✗ | ✗ | *Task-agnostic text* | Content embedding | Items |
| PC$^2$L [52] | ✓ | ✓ | Multimodal item features | Selected images | Explanation |
| COURIER [54] | ✓ | ✗ | Multimodal item features | Joint embedding | CTR |
| UniMP [46] | ✓ | ✗ | Multimodal item features | Joint embedding | Multi-task |
| VISUALLENS | ✓ | ✓ | *Task-agnostic images* | Spectrum | QA |

LLaMa models [53]. Retrieval-enhanced models, such as ReLLa, improve recommendation by retrieving relevant behavior sequences [27]. Instruction-tuning and graph augmentation approaches are explored in InstructRec [57], LLMRec [48], and LKPNR [15]. Jang et al. [20] use RLHF methods to improve personalizations as well. Recent advances in personalized conversation systems have utilized task-agnostic conversation logs to provide personalized answers [23, 16, 31, 29, 51]. However, these content-based recommendation approaches predominantly rely on textual data.

**Multimodal recommendation systems.** Multimodal recommendation systems [45, 54, 39] leverage multiple data types, such as text and images, to improve recommendation relevance and personalization. Before the LLM era, Lee and Abu-El-Haija [22] proposed systems for content-only video recommendation using similarity learning. PC$^2$L [52] develop an LLM model that provides multimodal explanations for recommendations. On modalities beyond image and text, MMRF [59] built a joint recommendation system integrating comments with video items [7, 8, 11, 58]. Rec-Former [25] and UniSRec [18] convert images to short captions to utilize text-only models. The current state-of-the-art image-text recommendation, UniMP [46], extended single-task multimodal personalization [17, 43, 49, 47] on multitask website-based shopping. However, most existing multimodal recommendation approaches rely on item-based user history, which is not always available. To address this limitation, we propose VISUALLENS, a novel framework that *leverages MLLMs to enable personalization using task-agnostic visual history*.

We discuss more related works on traditional recommendation and recommendation benchmarks in Appendix B.

## 3 Multimodal Task-Agnostic Recommendation

Consider a recommendation QA task, where the user asks a *recommendation question q*, and the recommender answers $q$ with a ranked list of candidate *items*. Good recommenders shall rank the items that the user is more likely to be interested in or willing to try early in the list.

In multimodal task-agnostic recommendation, the recommender is facilitated with a task-agnostic *visual history* $\mathcal{H}_u$ for each user $u$, which contains a series of photos, taken or posted by the user, not necessarily related to $q$. We state three assumptions to allow generalization. First, the photos may not be directly relevant to the question. Second, an image may not be associated with any candidate item, and even so, the candidate ID is not given. Third, a photo does not necessarily imply strong preferences. Figure 1 shows an example question, visual history, and candidates.

To simplify the problem, we assume a candidate retriever exists to retrieve all candidates that satisfy the user's question. Each candidate $s$ is represented with a $(x_s, \mathcal{I}_s)$ pair, where $x_s$ is the name and text descriptions, and $\mathcal{I}_s$ is an optional image set for the item.

Traditional recommendation setting considers two more types of signals. The first is a *task-specific* set of items in the candidate set, which captures user interest or at least user history. The second is a set of user-specific attributes such as the user's age, gender, and interests. This paper focuses on *task-agnostic* visual history and leaves integration of these traditional signals for future extensions.

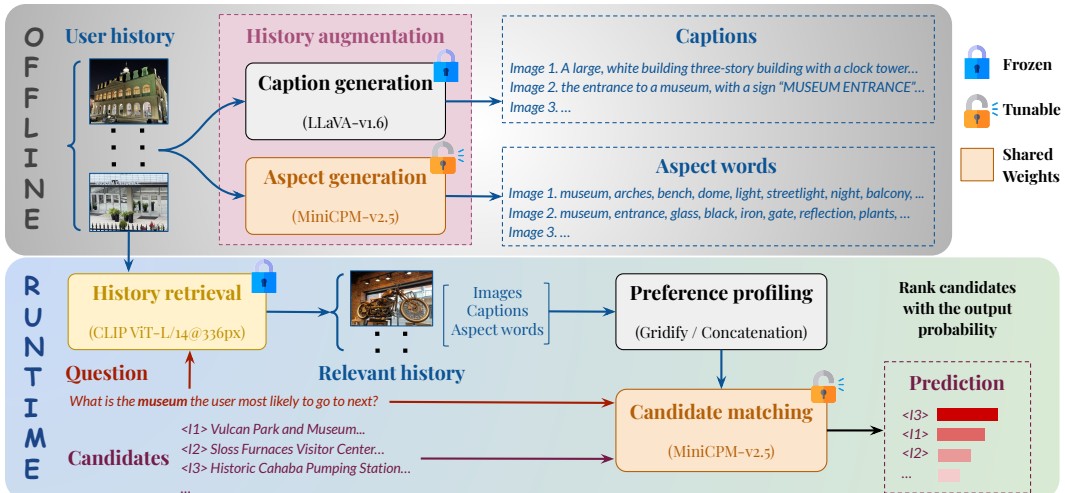

Figure 2: VISUALLENS inference pipeline: the offline process augments images with captions and aspect words to generate a spectrum user profile; the runtime recommendation process retrieves relevant images, generate query-specific user profile accordingly, and then predict candidate preferences.

# 4 VISUALLENS Framework

VISUALLENS framework contains two parts in inference (Figure 2): *offline user profile generation*, and *runtime recommendation*. Offline user profile generation (§4.1) augments each image in the visual history with *captions* and *aspect words*, and builds a spectrum user profile. Runtime recommendation (§4.2) answers a recommendation query $q$ in three steps. First, the *history retrieval* step retrieves only images relevant to $q$, since the visual history can be diverse and not all photos are relevant to every query. Second, the *preference profiling* step uses the retrieved images and their augmented captions and aspects to generate a query-specific profile of the user. Third, the *candidate matching* step matches the query-specific profile with each candidate, to generate the confidence score for each candidate for ranking. We discuss the joint training algorithm of VISUALLENS in §4.3.

## 4.1 Offline user profile generation

To build a user profile that retains rich visual content while offering clean, interpretable cues about user interests, we augment each image with a caption and a set of aspect words. Each image is thus represented as a spectrum triplet — (raw image, caption, aspect words) — ordered by decreasing information richness and increasing semantic clarity. Ablation results (Table 4) confirm that these augmentations improve recommendation performance.

**Image encoding.** Each image is encoded using the CLIP ViT-L/14@336px model [36], producing embeddings used for history retrieval at query time.

**Caption generation.** Captions are generated using a frozen LLaVA-v1.6-8B model [28], prompted to produce concise (≤30 words) and grounded descriptions to minimize hallucinations.

**Aspect word generation.** Aspect words are concise descriptors of key image attributes (*e.g.*, dome, balcony). We prompt the model to list relevant terms without constraining the count, allowing flexibility based on image complexity.

All modules are plug-and-play and can be replaced with stronger alternatives. While LLaVA-v1.6 occasionally produces irrelevant or generic aspect words (*e.g.*, blue, sky), we describe in Section 4.3.3 how joint finetuning improves their utility for recommendation tasks.

## 4.2 Runtime recommendation

**History retrieval.** Given a query $q$ and a user's visual history $\mathcal{H}_u$, VISUALLENS first retrieves images that are related to $q$, denoted by $\mathcal{I}_{u,q}$. We choose up to $w$ images to cap the number of images we

Table 2: Dataset statistics of Google Review-V and Yelp-V.

| Dataset | Train | Dev | Test | Categories | Avg. # of images | Avg. # of GT | Avg. # of candidates |
|---------|-------|-----|------|------------|------------------|--------------|----------------------|
| GR-V | 15.69M | 2K | 200K | 66 | 157.0 | 2.7 | 43.1 |
| Yelp-V | 4.12M | 2K | 100K | 35 | 263.6 | 8.2 | 66.7 |

process at runtime, and ensure that only the most contextually relevant images are retained for further processing, thereby reducing noise.

In general, we can use any image retrieval method such as DELG [5]. Here, we present a method for categorical recommendations such as *restaurants* and *museums*, popular in recommendation tasks. For each category $c$, we randomly select a set of candidate items in the category and average the visual embeddings of their images as the *image embedding of the category*, denoted by $\mathbf{v}_c$. Specifically, the category embedding is calculated as $\mathbf{v}_c = \frac{1}{n} \sum_{j=1}^{n} \mathbf{v}_c^{(j)}$, where $n$ is the number of candidates (see Appendix C for sensitivity), and $\mathbf{v}_c^{(j)}$ indicates the visual embedding of the $j$-th item image in category $c$. The retrieval step measures the cosine similarity between the visual embedding $\mathbf{v}_i$ of each image $i \in \mathcal{H}$ in the user's history and the image embedding $\mathbf{v}_c$ of the relevant category $c$. We then select the top-$w$ images based on the cosine similarity scores.

**Preference profiling.** Given a set of retrieved images $\mathcal{I}_{u,q}$, VISUALLENS then generates user's query-specific profile. A critical part of this step is image encoding. Even after retrieval, the number of images $w$ is still large. Most MLLMs allow context windows of limited sizes, constraining the number of images we can process. For example, for an input image of resolution $896 \times 896$, the PaliGemma model would generate an embedding of up to 4,096 tokens. A typical LLM with a context window of 8,192 tokens can take at most 2 images.

We propose to group relevant images $\mathcal{I}_{u,q}$ into a $d \times d$ grid, where $d^2 = w$, and treat all images in the grid as *a single image*. If we have retrieved fewer than $w$ images, we pad with a black background. Let $h$ be the maximum available resolution in a multimodal LLM. The *gridify* process $G$ takes the $d \times d$ grid and generates an image of fixed size $\mathbb{R}^{h \times h \times 3}$. Additionally, we number each image from 1 to $d^2$ to ensure the images are grounded to the corresponding caption and aspect words in the input to the candidate matching. We denote a user $u$'s profile on question $q$ by $(i_{u,q}, x_{u,q})$, where $i_{u,q}$ denotes the gridified image, and $x_{u,q}$ denotes the concatenated captions and aspect words of relevant images.

**Candidate matching.** Finally, VISUALLENS takes the query-specific user profile $(i_{u,q}, x_{u,q})$ and a set of candidates, each represented by $(x_s, \mathcal{I}_s)$, and generates the matching score for each candidate, which will then be used for ranking. This is achieved by prompting the multimodal candidate predictor, where we packed the user profile and candidates to the prompt through the image channel and the text channel separately (see the prompt template in Appendix D).

### 4.3 Iterative Refinement and Joint Training

VISUALLENS relies on LLMs for image encoding, caption generation, aspect word extraction, and profile–candidate matching. Current *off-the-shelf models perform suboptimally* for these tasks, so we apply continued pretraining and task-specific fine-tuning to enhance performance.

#### 4.3.1 Multi-image caption pretraining

To facilitate the model to ground to each grid faithfully, we perform a LoRA continual pretraining on dense captions. We adopt the dense captioning dataset **DOCCI** dataset [32], which contains over 15,000 images and their corresponding dense captions. Each time, we randomly sample $w$ images $\mathcal{I} = \{i_1, \cdots, i_w\}$ and their corresponding caption $\mathcal{C} = \{x_1, \cdots, x_w\}$, and then we construct a gridified input image $G(\mathcal{I})$ and a target output text description $T(\mathcal{C}) =$ "Image 1: $x_1, \cdots,$ Image w: $x_w$". Then we LoRA finetune the pretrained backbone model (*e.g.*, MiniCPM-V2.5) on all image-caption pairs $\{G(\mathcal{I}), T(\mathcal{C})\}$ so that the model is able to process the gridified user history grid by grid. We then use the continual pretrained model as the starting point to apply joint training described in §4.3.3.

### 4.3.2 Iterative aspect word refinement

Unlike image captioning, aspect word generation is not a standard multimodal task and lacks the extensive pretraining data. Hence, with zero-shot prompting, the generated aspect words have a large quality gap across different images, and the extracted aspects may not indicate user preferences.

To finetune aspect word generation, we first generate the training data. For each image $i$, we start from an initial set of aspect words, denoted as $\mathcal{W}_i^{(0)}$, which is generated by LLaVA-v1.6. In the $j^{\text{th}}$ round, we prompt a separate Llama-3.1 70B model with $\mathcal{W}_i^{(j-1)}$ candidates, and ground truths, and ask it to select *useful* aspect words that are helpful in ground truth prediction, which constitute $\mathcal{W}_i^{(j)}$. This refinement process continues for several rounds, and the iterations allow for converging extracted aspect words toward a more accurate and relevant subset. Empirically, we observe that the refinement converges after approximately 4 rounds, and denote the $4^{\text{th}}$ refined aspect word set $\mathcal{W}_i^{(4)}$ as $\mathcal{W}_i$, which serves as the training target.

The backbone model with parameter $\theta$ is finetuned to optimize the cross-entropy (CE) loss over all images $\mathcal{I}$,

$$\mathcal{L}_{\text{asp}} = \frac{1}{|\mathcal{I}|} \sum_{i \in \mathcal{I}} \text{CE}(\mathcal{W}_i, p_\theta(x_{\text{asp}}, i)), \tag{1}$$

where $x_{\text{asp}}$ is the prompt for aspect words generation.

### 4.3.3 Joint training of aspect word generation and candidate matching

To take advantage of multitask training in multimodal recommendation [46], we jointly train the aspect word generator and the candidate predictor on the backbone model. This joint training strategy allows the model to simultaneously learn to identify useful aspect words and make accurate predictions, thus improving overall performance.

The joint loss function balances aspect word generation and candidate matching with a weighting factor $\lambda$, where the candidate matching is optimized with binary cross-entropy (BCE) loss to handle multiple ground truth labels.

$$\mathcal{L}_{\text{pred}} = \frac{1}{N} \sum_{j=1}^{N} \text{BCE}(\mathcal{S}_j, p_\theta(x_{\text{pred},j}, i_{u_j,q_j})), \tag{2}$$

$$\mathcal{L}_{\text{joint}} = \mathcal{L}_{\text{asp}} + \lambda \mathcal{L}_{\text{pred}}, \tag{3}$$

where $u_j, q_j, \mathcal{S}_j$ is the user, the question, and the ground truth set of candidates of the $j$-th example. The text prompt $x_{\text{pred},j}$ consists of the question $q_j$, the text query-specific profiles $x_{u_j,q_j}$, and candidates. We LoRA finetune the model under the joint loss $\mathcal{L}_{\text{joint}}$.

## 5 Benchmarks and Experiments Setups

### 5.1 Benchmark creation

To the best of our knowledge, there is no existing benchmark [16, 31, 50, 44, 37] to evaluate personalization with task-agnostic visual history. We created two benchmarks, Google Review-V and Yelp-V, leveraging publicly available data from Google Local Review [24] and Yelp [2].

**User logs:** For each user in the two datasets, we take the list of reviews in chronological order. Each review is associated with a business *name, categories* and *description*. In Google Review-V, each review is associated with a few photos, used for image logs. Yelp-V does not associate a review with photos, so we randomly subsample one-third of the store profile pictures, such that different reviews for the same business can be associated with different images.

**Questions and visual history:** We consider a special type of questions, *category recommendations*, like *"Recommend a nearby museum"*. Such questions are both popular in real applications, and hard as there are many candidates satisfying the constraint. We remove small categories and most ambiguous categories, such as "place", and "spot".

Table 3: Hit rates and MRRs of VISUALLENS *vs.* multiple baselines on Google Review-V and Yelp-V. The result shows (a) VISUALLENS outperforms other baselines, though has a gap with the human oracle; (b) model size greatly affects the performance; (c) simply rank by rating is a worse design than the random baseline. Due to the large test set size (200K), an **MRR difference greater than 0.4 yields a $p$-value < 0.04**.

| | Modality | Size | Google Review-V | | | | Yelp-V | | | |
| --- | --- | --- | --- | --- | --- | --- | --- | --- | --- | --- |
| | | | Hit@1 | Hit@3 | Hit@10 | MRR | Hit@1 | Hit@3 | Hit@10 | MRR |
| *Naive baselines* | | | | | | | | | | |
| Random | - | - | 7.6 | 21.0 | 55.0 | 21.2 | 13.0 | 33.6 | 72.7 | 30.0 |
| Rank by rating | - | - | 3.9 | 15.8 | 55.5 | 17.7 | 8.7 | 28.0 | 72.3 | 25.9 |
| *Fine-tuned models* | | | | | | | | | | |
| UniMP [46] | T + I | 3B | 13.8 | 34.1 | 73.0 | 30.5 | 22.4 | 48.5 | 85.0 | 38.3 |
| Llama-3.1-8B-Instruct [30] | T | 8B | 15.8 | 36.3 | 77.2 | 32.9 | 24.1 | 52.2 | 88.5 | 39.6 |
| PaliGemma [3] | T + I | 3B | 13.0 | 32.0 | 70.1 | 28.4 | 20.8 | 46.7 | 82.0 | 37.5 |
| MiniCPM-V2.5 [56] | T + I | 8B | 16.1 | 36.4 | 78.4 | 33.2 | 24.8 | 53.0 | 89.3 | 40.3 |
| *Direct inference* | | | | | | | | | | |
| Llama-3.1-70B-Instruct [30] | T | 70B | 16.2 | 35.9 | 75.7 | 33.1 | 25.2 | 53.2 | 88.5 | 40.6 |
| GPT-4o [33] | T + I | - | 17.1 | 37.3 | 80.1 | 34.3 | 26.1 | 54.5 | 90.5 | 41.7 |
| *Our method* | | | | | | | | | | |
| VISUALLENS (PaliGemma) | T + I | 3B | 16.7 | 36.3 | 77.1 | 33.5 | 27.8 | 58.8 | 90.4 | 44.3 |
| VISUALLENS (MiniCPM-V2.5) | T + I | 8B | **18.5** | **38.9** | **82.3** | **35.4** | **28.3** | **59.1** | **91.0** | **44.9** |
| Human annotations | - | - | 22.0 | 45.0 | - | - | 36.0 | 66.0 | - | - |

For each review $r$ regarding a business of category $c$, we create a question asking to recommend businesses in category $c$. We take all (and only) photos in the reviews before $r$ to generate the visual history. We consider the visual history task-agnostic since the categories are highly diverse (see Figure 4), and the photos are quite diverse too (*e.g.,* a park photo to illustrate happiness mentioned in the review). We filter cases where the history is too short (<10) or does *not* contain the questioned category. By doing so, we ensure that the user history contains at least 10 relevant images (*i.e.*, from the same category as the query) for each example in both Google Review-V and Yelp-V.

**Candidates and ground truths:** For a review $r$ we take all reviews starting from $r$ to generate candidates and ground truths. To be realistic, we consider only nearby businesses of the same category as the candidate set, and the number of candidates is a random number in $[30, 100]$. Candidates that also appear in the user's future reviews are considered as ground truths. To avoid falling into a classification problem, we filter examples with only 1 ground truth in Google Review-V and fewer than 5 in Yelp-V.

**Summary:** Table 2 gives the benchmark statistics. The ratio of an average number of candidates and that of ground truths is 16:1 for Google Review-V and 8:1 for Yelp-V. By default, the train, dev, and test data have disjoint users. We discuss other splitting in Table 5 and more details in Appendix A.

## 5.2 Evaluation measures

We use two metrics to evaluate recommendation quality.

**Hit@k.** $\text{Hit@}k = \frac{1}{N} \sum_{i=1}^{N} \mathbb{1}[\text{rank}(r_i) \leq k]$ checks if any relevant item is within the top-$k$ ranked results, where $N$ is the number of examples, and $\text{rank}(r_i)$ is the rank of the first relevant item. We check Hit@3 (*e.g.*, voice recommendations) and Hit@10 (*e.g.*, on-screen recommendations).

**Mean Reciprocal Rank (MRR).** $\text{MRR} = \frac{1}{N} \sum_{i=1}^{N} \frac{1}{\text{rank}(r_i)}$ measures the ranking quality by averaging the reciprocal ranks of the first relevant item for each example. The MRR ranges from $1/S$ to 1, where $S$ is the number of candidates.

Additionally, we report wall-clock inference time in Appendix C to show the efficiency of VISUALLENS.

Table 4: Ablation study on PaliGemma. Different components of VISUALLENS model: joint training (Joint), iterative refinement (Iter), aspect words (Asp.), captions (Cap.), image embedding (Img.), and relevant image retrieval (Ret.). An **MRR difference greater than 0.4 yields a** $p$-**value < 0.04**.

| # | Representation | | | Ret. | Training | | Google Review-V | | | | Yelp-V | | | |
|---|---|---|---|---|---|---|---|---|---|---|---|---|---|---|
| | Asp. | Cap. | Img. | | Iter. | Joint | Hit@1 | Hit@3 | Hit@10 | MRR | Hit@1 | Hit@3 | Hit@10 | MRR |
| 1 | ✓ | ✓ | ✓ | ✓ | ✓ | ✓ | **16.7** | **36.3** | **77.1** | **33.5** | **27.8** | **58.8** | **90.4** | **44.3** |
| 2 | ✓ | ✓ | ✓ | ✓ | ✓ | | 16.1 | 35.8 | 76.2 | 33.0 | 27.2 | 57.9 | 88.9 | 43.3 |
| 3 | ✓ | ✓ | ✓ | ✓ | | | 15.7 | 35.2 | 75.4 | 32.5 | 26.9 | 57.5 | 88.2 | 42.9 |
| 4 | ✓ | | ✓ | ✓ | | | 15.2 | 34.7 | 74.2 | 31.9 | 25.8 | 55.3 | 86.1 | 41.2 |
| 5 | | ✓ | ✓ | ✓ | | | 14.8 | 33.9 | 73.0 | 31.2 | 25.0 | 53.9 | 84.9 | 40.4 |
| 6 | | | ✓ | ✓ | | | 13.5 | 32.5 | 71.9 | 29.6 | 22.0 | 48.2 | 83.6 | 38.8 |
| 7 | ✓ | ✓ | ✓ | | | | 11.5 | 27.9 | 67.3 | 25.9 | 20.1 | 45.7 | 81.7 | 36.8 |

## 5.3 Implementation and baselines

We ran VISUALLENS with two backbone models, a smaller 3B model PaliGemma and a larger 8B model MiniCPM-V2.5. For optimal performance, we selected $w = 64$ (112x112 each sub-image for PaliGemma, best in practice), corresponding to an $8 \times 8$ image grid, a candidate count of $n = 10$k, and a loss weighting factor of $\lambda = 2$ (Eqn. 3). We compared VISUALLENS with solutions below.

- *Baselines*: Randomly rank or select top-$k$ ratings.
- *Fine-tuned models*: We fine-tuned three state-of-the-art solutions: multimodal personalization model UniMP [46] (RedPajama 3B), with the adaptation to replace item images and attributes with image tokens in the visual history; Llama-3.1-8B-Instruct [30] with text-only user preference profiles; PaliGemma 3B and MiniCPM-V2.5 8B [56] with image profiles.
- *Direct inference*: We compared with two out-of-box models: Llama-3.1-70B-Instruct [30] with text-only preference profiles; GPT-4o [33] with multimodal profiles. To control the API cost, we subsample 1k instances from the test sets.
- *Human annotation*: Finally, we subsampled 50 examples from the test set for human annotation.

## 6 Results and Analysis

We conducted experiments to answer four questions:

**Q1**: Can we effectively leverage a user's visual history to improve personalization?

**Q2**: How does each element of VISUALLENS contribute to the recommendation quality?

**Q3**: Can VISUALLENS transfer across users, unseen categories, longer history, and new benchmark?

**Q4**: What is the robustness of VISUALLENS?

### 6.1 Recommendation effectiveness (Q1)

**VISUALLENS significantly outperforms baselines.** Our first observation from Table 3 is that all recommendation models that leverage the visual history significantly outperform baseline solutions on all metrics. In particular, the best version of VISUALLENS improves Hit@3 over *Random* by 18% on Google Review-V and by 26% on Yelp-V, and even more over *Rank by rating* (apparently, overall ratings do not cater to specific users). There is still a gap between VISUALLENS and human annotations, but comparing w. *Random*, it fills ~75% of the gaps for hit@3 on both datasets.

**VISUALLENS outperforms state-of-the-art solutions.** VISUALLENS outperforms UniMP with the same number of trainable parameters. With the 8B MiniCPM-V2.5 backbone, VISUALLENS outperforms MiniCPM-V2.5 itself by 2.5% on Hit@3 on Google Review-V, and by 6% on Yelp-V, and we observe a similar trend for the 3B models. Even comparing with significantly larger models, including Llama-3.1-70B-Instruct and GPT-4o without fine-tuning, VISUALLENS 7B improves by 1.6-5.6% on Hit@3.

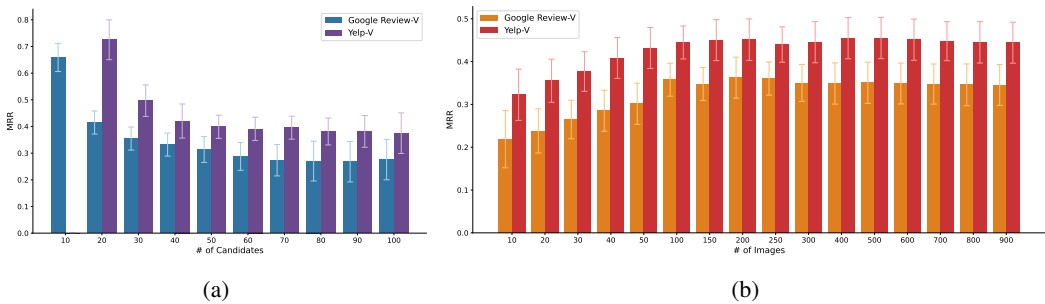

|       (a)       |       (b)       |

Figure 3: (a) MRR distribution over number of candidates, (b) MRR distribution over number of images. Both are on the User ID test set. We find (1) MRR converges when number of candidates exceeds 50; (2) MRR increases and flattens after reaching ~100 images.

Table 5: Transferability: MRR of VɪsᴜᴀʟLᴇɴs models to different test setups. LongHis: train until a certain timestamp and test afterwards. Category: held-out a set of categories for testing per user. Use ID: train and test set share no common user ID.

|  | Google Review-V | | | Yelp-V | | |
| --- | --- | --- | --- | --- | --- | --- |
|  | LongHis | Category | User ID | LongHis | Category | User ID |
| VɪsᴜᴀʟLᴇɴs (PaliGemma) | 35.9 | 34.9 | 33.5 | 46.6 | 45.2 | 44.3 |
| VɪsᴜᴀʟLᴇɴs (MiniCPM-V2.5) | 38.0 | 37.1 | 35.4 | 47.2 | 45.5 | 44.9 |

## 6.2 Ablation studies (Q2)

We evaluate the usefulness of each component in VɪsᴜᴀʟLᴇɴs in Table 4, with PaliGemma as the backbone. We find history retrieval significantly improves the results, and can reduce Hit@3 by 7% on Google Review-V and by 12% on Yelp-V (#7 vs. #3). Besides, all three representations of the images (embedding, caption, aspects) play an important role. Removing captions and aspect words can reduce Hit@3 by 3% on Google Review-V and by 9% on Yelp-V, even without fine-tuning (#6 vs. #3). Between the two, aspect words play a more important role than captions (#5 vs. #4). Moreover, both iterative training and joint multi-task training improve the recommendation quality. Removing both of them lowers Hit@3 by 1%+ on both data sets (#3 vs #1).

## 6.3 Transferability (Q3)

We tested transferability over users (default setting), over categories, and over different (longer) history lengths. Table 5 compares the MRR of VɪsᴜᴀʟLᴇɴs on both benchmarks, showing good transferability. MRR is highest when applied to longer history, with the effectiveness of history retrieval. Transferability is higher across categories than across users, both demonstrating the promise of leveraging task-agnostic signals, and illustrating slightly more challenges to transfer between users of different interest patterns. We also present the generalization results across benchmarks in Appendix C.

## 6.4 Robustness and qualitative analysis (Q4)

We conducted several robustness tests as follows.

**Candidate count:** Figure 3a shows that as the number of candidates grows, the recommendation becomes harder and MRR gets lower. However, when the number of candidates exceeds 50, MRR converges.

**Image count in user history:** Figure 3b shows that as the history grows, MRR increases and flattens after reaching ~100 images. This trend is related to our grid size $8 \times 8$, as smaller than 64 images will not leverage all spaces in the grid. On the other hand, flat MRR after 100 images shows robustness against history noises with the retrieval step.

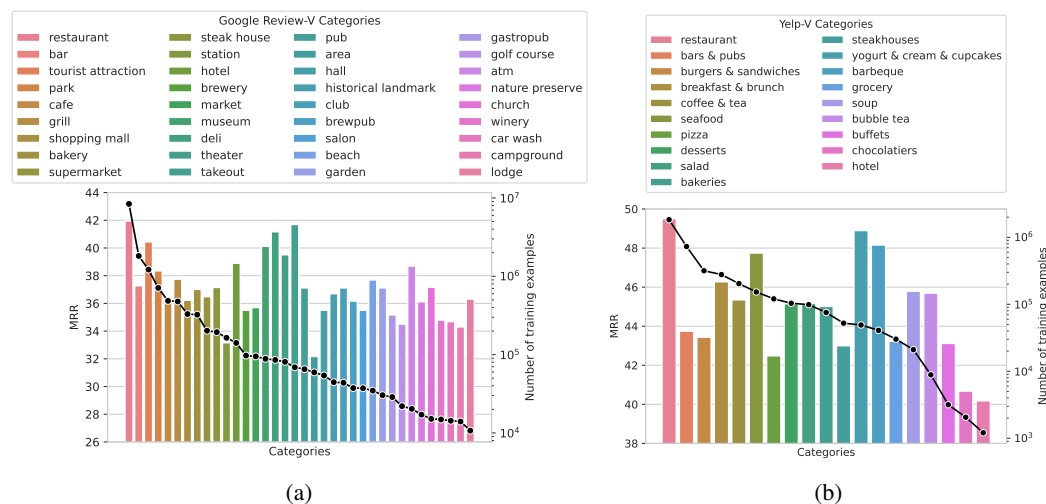

Figure 4: (a) MRR distribution over categories on Google Review-V, (b) MRR distribution over categories on Yelp-V. We find (1) the performance per category is loosely correlated with number of training data; (2) when a category is more general and less ambiguous, the performance on the category is better.

**Category distribution:** Figure 4 plots MRR for different categories. There are a few factors that can affect recommendation quality. First and foremost, ambiguous categories like "area", "station", and "market" get the lowest MRR in Google Review-V. Second, general categories (*e.g.*, "museum", "hotel") with bigger category sizes obtain higher MRR than specific ones (*e.g.*, "historical landmark"). Third, transfer learning happens between neighbor categories; for example, "deli" and "takeout" achieve the top-2 and top-3 performance with less training data, since they are similar to the largest category "restaurant". We provide more qualitative analysis in Appendix C.

## 7 Conclusion and Discussion

In this paper, we proposed a novel approach VISUALLENS for personalized recommendation using task-agnostic visual history. We advanced MLLMs to extract spectrum signals from images to serve as user profile. We created two benchmarks, Google Review-V and Yelp-V, to evaluate VISUALLENS, affirming the efficacy and robustness of VISUALLENS.

VISUALLENS offers a promising first step toward MLLM recommendation systems that leverage task-agnostic visual history. Several future directions remain. First, we could integrate VISUALLENS with additional data, such as image timestamps, locations, recognized fine-grained entities (*e.g.*, specific products), and user profiles. Second, we aim to extend the recommendation problems explored here to encompass broader QA tasks. Finally, we plan to investigate privacy-preserving techniques, such as federated learning, during training.

## Social Impact

The ability to model user preferences from visual histories raises important considerations around privacy, consent, and data usage. While our work uses publicly available and anonymized images, real-world deployments would need to carefully address how user data is collected, stored, and interpreted, especially when recommendations extend beyond a single domain. There is also potential for reinforcing biases or overfitting to superficial cues (*e.g.*, location, aesthetics) that may not reflect deeper user intent. We encourage future research to incorporate fairness auditing, privacy-preserving training, and transparency mechanisms to ensure responsible use of multimodal personalization systems.

## Acknowledgements

The robot icon used in Figure 1 is from https://www.flaticon.com/free-icon/robot_2432846. The padlocks icons used in Figure 2 are from https://www.flaticon.com/free-icon/lock_996365 and https://www.flaticon.com/free-icon/padlock_535143. The illustration images in Figures 1, 2 are from DOCCI [32].

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

# Appendix

# A  Benchmark Creation Details

## A.1  Algorithm for candidate generation

We list the complete candidate and ground truth sets generation algorithm in Algorithm 1, where the function nearest($\mathcal{G}$, loc, $m$) returns the $m$ nearest businesses of around certain location loc based on the graph $\mathcal{G}$. The function unique_name($\mathcal{S}$) removes the businesses in the set $\mathcal{S}$ with redundant names.

In both Google Review-V and Yelp-V, $rc_{\min} = 30$, $rc_{\max} = 100$, $fc_{\min} = 10$.

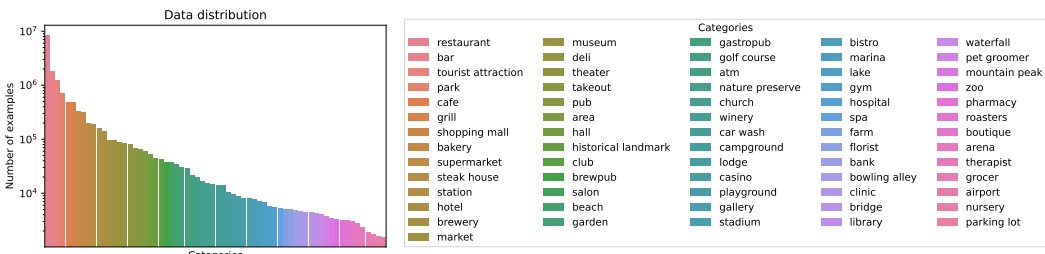

Figure 5: The Google Review-Vision (Google Review-V) training data consists of 66 categories.

**Algorithm 1** Candidate and Ground Truth Sets Generation

---

**Input:** Business geographics graph $\mathcal{G} = (\mathcal{V}, \mathcal{E})$, Visit set $\mathcal{B}$ of a user, Category $c$, Minimum random candidate count $rc_{\min}$, Maximum random candidate count $rc_{\max}$, Minimum final candidate count $fc_{\min}$, Minimum ground truth count $gtc_{\min}$.
**Output:** Candidate sets $\mathcal{S}_{1..n}^{\text{CD}}$, Ground truth sets $\mathcal{S}_{1..n}^{\text{GT}}$.
*# Initialization*
**for** each business $b \in \mathcal{B}$ **do**
    flag $f_b \leftarrow unselected$
**end for**
$count \leftarrow 0$
*# Main algorithm*
**for** each business $b \in \mathcal{B}$ **do**
    **if** $b$ is in category $c$ **and** $f_b = unselected$ **then**
        $count \leftarrow count + 1$
        Candidate count $m \leftarrow \text{random}(rc_{\min}, rc_{\max})$
        Candidate set $\mathcal{S}_{count}^{\text{CD}} \leftarrow \text{nearest}(\mathcal{G}, b_{\text{loc}}, m)$
        Candidate set $\mathcal{S}_{count}^{\text{CD}} \leftarrow \text{unique\_name}(\mathcal{S}_{count}^{\text{CD}})$
        Ground truth set $\mathcal{S}_{count}^{\text{GT}} \leftarrow \mathcal{S}_{count}^{\text{CD}} \cap \mathcal{B}$
        **if** $|\mathcal{S}_{count}^{\text{CD}}| < fc_{\min}$ or $|\mathcal{S}_{count}^{\text{GT}}| < gtc_{\min}$ **then**
            $count \leftarrow count - 1$
            **continue**
        **end if**
        **for** each business $b' \in \mathcal{S}_{count}^{\text{GT}}$ **do**
            flag $f_{b'} \leftarrow selected$
        **end for**
    **end if**
**end for**
**return** $(\mathcal{S}_{1..count}^{\text{CD}}, \mathcal{S}_{1..count}^{\text{GT}})$

---

## A.2 Processing and filtering

For Google Review-V and Yelp-V, the categories are selected as the last word of the annotated tags in the business. However, some category requires multiple words to express its meaning, such as "`tourist attraction`", "`steak house`", "`historical landmark`", "`nature preserve`", etc. We select and keep these multi-word categories.

In Google Review-V, we remove smaller categories with less than 10k occurrence in the dataset. Then, we remove ambiguous or non-differentiable categories, including "`shop`", "`store`", "`complex`", "`service`", "`company`", "`supplier`", "`caterer`", "`agency`", "`center`", "`organization`", "`attraction`", "`house`", "`mall`", "`landmark`", "`wash`", "`course`", "`preserve`", "`alley`", "`groomer`", "`field`", "`peak`", "`venue`", "`delivery`", "`dealer`", "`lounge`", "`office`", "`arcade`", "`court`", "`spot`", "`stop`", "`maintenance`", "`trainer`", "`wholesaler`", "`planner`", "`place`", "`facility`", "`school`", "`stand`", "`range`", "`consultant`", "`designer`", "`veterinarian`", "`ground`", "`contractor`", "`manufacturer`", "`studio`", "`point`", "`lot`".

In Yelp-V, as the category is very centralized, we remove smaller categories with less than 50k occurrence in the dataset. Then, we remove ambiguous or non-differentiable categories, including "`planning`", "`nightlife`", "`services`", "`wings`", "`arts`", "`dogs`", "`tacos`", "`caribbean`", "`beer`", "`spirits`", "`wine`", "`venues`", "`fusion`", "`entertainment`", "`southern`", "`spaces`", "`lounges`", "`breweries`", "`shopping`", "`smoothies`", "`flavor`", "`plates`", "`eastern`", "`tex-mex`", "`shop`", "`noodles`", "`markets`", "`market`", "`donuts`", "`gelato`", "`sum`", "`veggies`", "`fruits`", "`trucks`", "`bagels`", "`cheesesteaks`", "`clubs`", "`cuban`", "`ramen`", "`life`", "`roasteries`", "`stands`", "`brewpubs`", "`gluten-free`", "`gardens`", "`travel`".

## A.3 Training data distribution

We list the training data distribution of Google Review-V in Figure 5.

## B  More Related Works

### B.1  Traditional recommendation methods.

Before the LLM era, there are also a line of work for recommendation with other networks, smaller language models, or ensembling methods. For example, Sun et al. [40] uses a bidirectional Transformer to model the item sequence. Tang and Wang [42] uses CNN to model user preference with convolutional filters. Gu et al. [14] and Yang et al. [55] uses ensembling methods for recommendation.

### B.2  Multimodal large language models.

Multimodal LLMs are becoming increasingly powerful in processing images and videos and in generating human-like text, exemplified by models like GPT-4 family [33], Claude 3.5 family [1], Gemini and PaliGemma [13, 3], LLaVA model family [28], and Llama 3 Vision models [30]. However, they still suffer from a strong language prior [9, 35], generalization [60] and hallucinations [10], or require heavy text transcription [61]. The extent to which they can understand user history, in analogy to LLM recommendation systems, is still unclear.

### B.3  Datasets and evaluating multimodal recommendation.

The development of multimodal recommendation systems has been facilitated by the availability of diverse datasets that incorporate various types of data, including text, images, and user interaction histories. Notable datasets in this domain include MovieLens-1M [16], which is extensively used for movie recommendations, and the Amazon dataset [31], which includes user reviews and metadata for product recommendations. For news and books recommendations, the MIND [50] and Goodreads [44] datasets offer insights into user preferences in news and literature. Recently, LaMP [37] introduces 7 text classification and text generation tasks across long contexts to evaluate LLM's personalization capablity. In the context of outdoor activities, datasets such as Google Local Data [24, 52] and Yelp [2] are crucial. These datasets not only provide textual reviews but also include user ratings and geospatial data, which are essential for recommending local businesses and services. These benchmarks utilize the traditional recommendation systems that focus on ranking user preferences based on historical IDs. Instead, we introduce new visual history benchmarks Google Review-V and Yelp-V, using data from Google Local and Yelp, to evaluate the multimodal recommendation systems under a more realistic visual history for task-agnostic setups.

## C  More Analysis

### C.1  Sensitivity of random sampling for candidate embedding

We show that the recommendation results are not sensitive to the random sampling of images in candidate embedding. Table 6 below shows that the robustness against sampling randomness: 1) the standard deviation is <0.1 when we do 5 runs of sampling; 2) as we reduce the size of the samples to 1K and 100, the MRR stays similar, with <0.5 difference.

Table 6: MRR and its standard deviation (5 runs) of VISUALLENS (PaliGemma) on different sampling sizes.

| $n$ | Google Review-V | Yelp-V |
| --- | --- | --- |
| 10k | 33.5 ± 0.1 | 44.3 ± 0.0 |
| 1k | 33.4 ± 0.1 | 44.3 ± 0.1 |
| 100 | 33.0 ± 0.2 | 44.1 ± 0.1 |

### C.2  Generalization between Google Review-V and Yelp-V

We report the generalization quality between Google Review-V and Yelp-V. Table 7 below shows that cross-source recommendation reduces MRR by 2-4% respectively *vs.* in-domain, in an acceptable range, highlighting the transferability from the shared feature in the embedding space. Besides,

transferring from Google Review-V to Yelp-V is still better than the best baseline model in-domain (40.2), while the other side is much worse (33.2). This is as expected, since Google Review-V has much more categories and much lower photo quality.

Table 7: MRR of VISUALLENS (PaliGemma) on in-domain test and cross-domain transfer.

| Train data | Test data | |
|---|---|---|
| | Google Review-V | Yelp-V |
| Google Review-V | 33.5 | 41.9 |
| Yelp-V | 29.4 | 44.3 |

## C.3 Latency evaluation

To evaluate the latency of VISUALLENS, we measured inference wall-clock time per example on a single NVIDIA L40S GPU. The results in Table 8 demonstrate that the joint image representation learned by VISUALLENS is more efficient and effective than the interleaved multimodal representation used in UniMP, reducing latency by ≈0.5 sec.

Table 8: Average wall-clock inference time of VISUALLENS compared with UniMP.

| Model | Google Review-V | Yelp-V |
|---|---|---|
| UniMP (3B) | 5.82s | 4.92s |
| VISUALLENS (3B) | 5.29s | 4.55s |
| VISUALLENS (8B) | 7.31s | 6.64s |

## C.4 Qualitative results

We show the input and output of a positive example in Figure 6 and Figure 7. We show the input and output of a negative example in Figure 8 and Figure 9. Please note that due to image licensing restrictions, we are only able to provide the image names in JPG format. For more details, you can download the image data from https://business.yelp.com/data/resources/open-dataset/.

## D   Prompt Template

We list the prompt template for aspect word generation in Figure 10, and the prompt template for candidate matching in Figure 11. The model will fill in category information in the [[Category]] slot and predict the answer at the [[Answer]] slot.

## E   Additional Implementation Details

We present the hyperparameters for training VISUALLENS in PaliGemma and MiniCPM-V2.5 in Table 9 and Table 10. Note that since there are at most 100 candidates, we add special tokens <I1>, ..., <I100> as identifier of the candidates in the vocab of the models. By doing so, we can predict the rank by taking the probability of only those special tokens in the first output token.

## F   Licenses

We list the licenses involved in this work as follows.

- Our usage of Google Local Data is under the license of Google Maps Platform Terms of Service.
- Our usage of Yelp Data is under Yelp Dataset License.

**Question:** *What is the seafood the user most likely to go to next?*

**Images:** vJqN7cQXBVKHJJPXLZhoiQ.jpg, r4AfL_MgfmF3kmPMFs8TFg.jpg, UKMOX-Zvg6jMFNw7q65oEA.jpg, wPn4exXYYNKP3lzSqyvlLg.jpg, 2ArnN2RJkdSRsiTfTdEK5A.jpg, pRwlK1-CwGNcuaj68ajV_A.jpg, tX20F76pfX54eOh1kkznsg.jpg, YQiidvr8-Pb3Bl7PS1EvTw.jpg, 0nuzLrJuSMfbuwK_eu20CQ.jpg, TXGOYaPCse3jSjNb22qBbg.jpg, CNF90JgRzw8OSDCtUWOwJQ.jpg, H0lVV7AqqNHqIxg5W-7VqA.jpg, hC5YoVNzvbsSTIBzYUw6gQ.jpg, 2t1gOf8J6RdGAN08YtzK6A.jpg, wyomfq33QG7Lcsm2I1keYg.jpg, xBEVh2jWRaHFo-GVDkdYww.jpg, 8rXB-RWfztDFTUXd4v0AjQ.jpg, 7jT9OQ6iWhysKmfKtSLjSw.jpg, dMq7TX-_HAZevi0ec2_1_w.jpg, h2vbvkl-qVtFykWSnXMA6A.jpg, MQgymZxJgwvZi9ceTKhJqw.jpg, BYGg4bPM9RyOlcpbj_pdPA.jpg, tfxriiwBgyz-kTYhWygpCg.jpg, Ara9kbHnH_kiwWf03Nr-dA.jpg, yFE4KT49_8uY5CRD4N8ZYQ.jpg, ANXYMnTSlIdcSSxBTaeuyg.jpg, S3jQKlAJqGC4TU0gDK66Hw.jpg, spTRlxA-jfJWTyZTw2cgVg.jpg, qJoah_eRYngBqr9loubCbA.jpg, MQuNacSddcwXFtoblQ1GkQ.jpg, EhkCMYoscdfMpGubTBfl3A.jpg, y4r4dgzZfPo4q_5YAiGHvg.jpg, jxKIXDjCqAsqyakhCUC3Lw.jpg, 2W7reraU4LA0XrP78y64sw.jpg, FhVHa7bRVrzMda_Dm1wnTg.jpg, KWzV50vfln0_ag5qs989OQ.jpg, Mc25dRycrpy2SHGihHOW0g.jpg, FmT3RYVQY6Rkm5MlYt1EAQ.jpg, zIRzC3yzhcSHO8VLTemtFg.jpg, _hkeputL9COjS_sXrRDeIg.jpg, 0oyPpSHInW-lnnToM-xeXQ.jpg, 3Bkrs4_XOysf5EBx7tE_lg.jpg, deQ74JW91K-St4RgULT-nQ.jpg, WPKSSCdLPteqrwpeS9hZvA.jpg, 9biNDFQDY8r9BlasGKk2GA.jpg, wi4Tm0FMuQ2okQwYtHPyWw.jpg, tsjxColf4XZmBfq-B7WJLQ.jpg, Y0D5OiZPVikxP3nXzSE19Q.jpg, YQuc3-t3p2lnQyaWEE0xGQ.jpg, ezf9bs1GI44t8XkBiYOHkQ.jpg, fOe1E9JsDCGGxTuJaxJUQg.jpg, a8cVlIBv2ztpNq-zU93Ghw.jpg, RfO6Kr6H0RrA5T_L50B1mQ.jpg, OkelD8PM3aYV_oiT_OOfPQ.jpg, Q6ZABL2V_WkUcjZAQS48Cg.jpg, VaOy8CE5EO0-ZArdyQQkIA.jpg, dLvdOgh417CZBqO3NUDriQ.jpg, 8tX8MDZ7nNBsbOw4M7Qfiw.jpg, QIYfwuQc-Pw2erU501ipYQ.jpg, yvvlCHgMZIa8Vg9UlIQ9Iw.jpg, mWjD72pB7TnoSYCOW_EYQw.jpg, TK5g5F19WJZuTsXxf77XVg.jpg, NPsJmoYIdvoRzTv4l-jjmg.jpg, 2MlS2-V5AJcVfHInXUxCDg.jpg, UlJkD0OHYsXjPbBUXMIvCg.jpg, oowr6hZezzRrAf15FfNaRQ.jpg, ifEeKdHzwpI4He54LylxFw.jpg, S5zpR2aJD5opF3TSyr5GHg.jpg, f0jBRH-6c0C7LKRMZ_7CHA.jpg, iIpBj0_0F-FnGu5byUXHHw.jpg

**Candidates:**

| | |
|---|---|
| <I1> Stoney River Steakhouse and Grill | <I2> JWB Grill |
| <I3> Morton's The Steakhouse | <I4> The Hampton Social - Nashville |
| <I5> Char Restaurant | <I6> Batter'd & Fried Boston Seafood House |
| <I7> Captain D's | <I8> Willie B's Kitchen & Lounge |
| <I9> Ruth's Chris Steak House | <I10> Bob's Steak & Chop House |
| <I11> Fish & Co | <I12> Poke Bros |
| <I13> Rudie's Seafood & Sausage | <I14> Far East Nashville |
| <I15> Demos' Restaurant | <I16> Sehrt Seafood Company |
| <I17> Firebirds Wood Fired Grill | <I18> Reyes Mexican Grill & Mariscos |
| <I19> East Side Fish | <I20> Crab Fever |
| <I21> Sitar Indian Restaurant | <I22> Carrabba's Italian Grill |
| <I23> Jeff Ruby's Steakhouse- Nashvill | <I24> Fleming's Prime Steakhouse & Wine Bar |
| <I25> Amerigo Italian Restaurant | <I26> The River House |
| <I27> Etch | <I28> Fannie Mae's |
| <I29> The Hook | <I30> Green Hills Grille |
| <I31> Pemrose | <I32> The Southern Steak & Oyster |
| <I33> Oak Steakhouse Nashville | <I34> El Tapatio #2 |
| <I35> Red Perch | <I36> Marsh House |
| <I37> Sushi 88 | <I38> RuSan's Sushi and Seafood |
| <I39> Skull's Rainbow Room | <I40> No 1 Chinese |
| <I41> Little Octopus | <I42> The Cheesecake Factory - Nashville |
| <I43> Juicy Seafood | <I44> House of Cards |
| <I45> Joe's Crab Shack | <I46> Henrietta Red |
| <I47> Urban Grub | <I48> Bombay Palace Restaurant |
| <I49> The Gumbo Bros | <I50> The Diner |
| <I51> Redlands Grill - Nashville | <I52> The Green Pheasant |
| <I53> Nomzilla! Sushi Et Cetera | <I54> The Smiling Elephant |
| <I55> Sperry's Restaurant | <I56> Louie's Wine Dive |
| <I57> Eddie V's Prime Seafood | <I58> The Optimist - Nashville |
| <I59> The Silly Goose | <I60> Cousins Maine Lobster - Nashville |
| <I61> Jimmy Kelly's Steakhouse | <I62> J. Alexander's Restaurant |
| <I63> 360 Wine Bar Bistro | <I64> The Palm - Nashville |
| <I65> Blue Moon Waterfront Grille | <I66> Bourbon Steak by Michael Mina |
| <I67> Boston Commons | <I68> White Limozeen |
| <I69> Saltine Fine Food and Drink | <I70> Seafood Sensation |
| <I71> Siam Cafe | <I72> The Continental |
| <I73> South Street | |

Figure 6: A successful example of VISUALLENS on Hit@1. We list the input question, images and candidates.

- PaliGemma models are under a custom license the Gemma Terms of Use (https://ai.google.dev/gemma/terms).

- MiniCPM-V2.5 is under Apache License 2.0

- Our usage of OpenAI's models for prompting is under the license of OpenAI"s Terms of Service.

```
                              ┌─── Output ───┐

Candidate Ranking:
<I66> Bourbon Steak by Michael Mina      <I33> Oak Steakhouse Nashville
<I61> Jimmy Kelly's Steakhouse           <I23> Jeff Ruby's Steakhouse- Nashville
<I9> Ruth's Chris Steak House            <I24> Fleming's Prime Steakhouse & Wine Bar
<I10> Bob's Steak & Chop House           <I3> Morton's The Steakhouse
<I64> The Palm - Nashville               <I72> The Continental
<I47> Urban Grub                         <I57> Eddie V's Prime Seafood
<I58> The Optimist - Nashville           <I32> The Southern Steak & Oyster
<I17> Firebirds Wood Fired Grill         <I1> Stoney River Steakhouse and Grill
<I5> Char Restaurant                     <I51> Redlands Grill - Nashville
<I36> Marsh House                        <I69> Saltine Fine Food and Drink
<I26> The River House                    <I44> House of Cards
<I62> J. Alexander's Restaurant          <I46> Henrietta Red
<I55> Sperry's Restaurant                <I2> JWB Grill
<I4> The Hampton Social - Nashville      <I6> Batter'd & Fried Boston Seafood House
<I13> Rudie's Seafood & Sausage          <I16> Sehrt Seafood Company
<I60> Cousins Maine Lobster - Nashville  <I70> Seafood Sensation
<I29> The Hook                           <I35> Red Perch
<I43> Juicy Seafood                      <I45> Joe's Crab Shack
<I49> The Gumbo Bros                     <I11> Fish & Co
<I38> RuSan's Sushi and Seafood          <I20> Crab Fever
<I41> Little Octopus                     <I65> Blue Moon Waterfront Grille
<I27> Etch                               <I39> Skull's Rainbow Room
<I68> White Limozeen                     <I56> Louie's Wine Dive
<I63> 360 Wine Bar Bistro                <I30> Green Hills Grille
<I50> The Diner                          <I73> South Street
<I31> Pemrose                            <I12> Poke Bros
<I52> The Green Pheasant                 <I67> Boston Commons
<I59> The Silly Goose                    <I28> Fannie Mae's
<I8> Willie B's Kitchen & Lounge         <I37> Sushi 88
<I53> Nomzilla! Sushi Et Cetera          <I15> Demos' Restaurant
<I18> Reyes Mexican Grill & Mariscos     <I19> East Side Fish
<I22> Carrabba's Italian Grill           <I25> Amerigo Italian Restaurant
<I34> El Tapatio #2                       <I42> The Cheesecake Factory - Nashville
<I54> The Smiling Elephant                <I71> Siam Cafe
<I7> Captain D's                         <I14> Far East Nashville
<I21> Sitar Indian Restaurant            <I40> No 1 Chinese
<I48> Bombay Palace Restaurant

                          ┌─── Ground Truth ───┐

<I12> Poke Bros
<I22> Carrabba's Italian Grill
<I33> Oak Steakhouse Nashville
<I57> Eddie V's Prime Seafood
<I61> Jimmy Kelly's Steakhouse
<I66> Bourbon Steak by Michael Mina
<I68> White Limozeen
```

Figure 7: A successful example of VISUALLENS on Hit@1. We list the output candidate ranking and ground truth. The ranking follows the same left-right order as the input candidates.

Table 9: Hyperparameters for training VISUALLENS with PaliGemma Backbone.

| Hyperparameters for training on PaliGemma | |
| --- | --- |
| Parameter Size | 3B |
| Image Resolution | $896 \times 896$ |
| Number of Image Tokens | 4096 |
| Hidden Dimension Size | 2048 |
| LoRA Rank | 16 |
| LoRA $\alpha$ | 16 |
| LoRA dropout | 0.1 |
| GPU | $8 \times$ NVIDIA H100 |
| Batch Size | 8 |
| Gradient Accumulation Steps | 8 |
| Warmup Steps | 200 |
| Learning Rate | 1e-3 |

**Question:** *What is the* ***pizza*** *the user most likely to go to next?*

**Images:** hhRkpdongqf8m5EE10YQRQ.jpg, 9ToJiDVDJy5pUnSoMNGuBg.jpg, KFx5LzegrM_8TFwliDk9aQ.jpg,
742SAAE7x0gmDRGsKHQ4UA.jpg, 6q5oq-bpGqEQ6jEnwy-myA.jpg, KC6EHtjGVe-Dvx8v5R3tEA.jpg,
VSJRG9TWrNukgVuk9MS1cQ.jpg, 5pYM0N5hx0el5R1C1DCYew.jpg, UPQQfA1mrMEJSYP18_QjBw.jpg,
9z4Pu_zbWb-W7Y_j-7H1Yg.jpg, Ido4zXZ7KOQeRY5hNuBAsg.jpg, Rmtu1zLSG68i1d4JT62CQQ.jpg,
ItWufsYqPm0JQxoohSTiUA.jpg, dJtqcFC8PRud3yk8s1TvvQ.jpg, c586W6qreQigu6EgltB3Ow.jpg,
c7g6ciD8KYu3vb6Cw0DGCA.jpg, 3Ob64cs6HgG0CCUY-ppj4A.jpg, rOzJH8qZYacB2Xa-AbzE9A.jpg,
mO4d4KAlxdvNdUGu0av7kQ.jpg, xou5bMV0GxYrVUU3tCDDBg.jpg, uzbaUWOJnC3f9V35pJUn-A.jpg,
QVGyJPteap1ueoMz8o5JsA.jpg, cTeR6ZQh-KQgcaO1jXMbpQ.jpg, 66YOiOkDwsvFYkCiOBa3KQ.jpg,
By97Yxw5e4Ab5DfpEMQ6uw.jpg, t_Vr-IjXTgE6PZg7qUpafA.jpg, EbsYH1VMwXtQuLXwdFw1Zw.jpg,
-MvxNZN4RIkzNRBc003DwQ.jpg, 0nCG8FvsObbrpYulUmnuTQ.jpg, 5Zg5zYaGlpkApTIVFTqeGQ.jpg,
_AFOqfRIUIwyJM1FSzI0EA.jpg, Hi3drxwhhP_qvHeC8VbGAg.jpg, RHyRk_8lHMzEtlh2-TLdrQ.jpg,
z_xa3u4wx4NHjW8eoZHUvw.jpg, G9B1SMaiYILP8AjC4pTQ1A.jpg, aEgf8X4R4bJDuAE_kYjtKw.jpg,
R96I_J2M81SYz0WdXQe_oA.jpg, x9uHc7Y9Bzy7KkwDeJx2Kw.jpg, uQ5xcTpP2JAsUOohT8DI2g.jpg,
UseElxL5U7wYbmlf9Az7kQ.jpg, QySJ5zZhePEuQLrixs5O7w.jpg, HqJ_3OzutHD-yr2UajmA8w.jpg,
i8JBFOQ907Akk-VutW5gSA.jpg, ZELwmcr7KW1MNqMkAN9E2Q.jpg, BgVWrsi78V-BxiLxaj9PGA.jpg,
zoeOP-t0QR_1b4iDC6j2HQ.jpg, SnkcEBSZMuQCpbWCjVkchw.jpg, VeBxRrP3g2T3lJD_dOTxGQ.jpg,
-mIOvTzcHsjr1drlapEITA.jpg, hs4OYvrita3-pM4ags835A.jpg, HhoyiXcspjVILME9pGG00g.jpg,
FHCf0kMZMzO9DCHC26QjOQ.jpg, 3b5QJj8GQA8g5jipS-WWjw.jpg, IGVKYExRpkDoQBWupcIRDQ.jpg,
HDELx9GYX6iUvtZxQ8ctcA.jpg, aws74_-4V-xybaXhOp7KxQ.jpg, BUlr9EAkvHjA68HGkTjzpQ.jpg,
33HfUzGpm0pojsul6xrnnA.jpg, hcztlNNBk1OpGj51XBbfRg.jpg, uWj0ff2rembupvVUJnfyWQ.jpg,
1lxJBNjg7ZzZhNUWUpqbVA.jpg, wNBmdFh-m-owSFOVdlz3xw.jpg, V1sHwJCt_1-WRl3_bgnuCA.jpg,
eCi5nuodtwhbMhNzBDCXvw.jpg, YqOF3R95qupsBzlaDBSW_g.jpg, vwFLRRbHDAq8qI4d0_hslg.jpg,
N0aSm4quN2s-XUxz2LnhxA.jpg, qz3bDurS9JjWX6v2_pJBKA.jpg, _ykibX3snmvSYQctujwEWg.jpg,
OcF6jXTNLUC20WhUVSfcpg.jpg, 3jGXj-oz5VtbqjIm9L93xA.jpg, Tsnd-cwmKWNfdkpcI-U8yw.jpg,
QaysW7lx3PpfwuoXuYH-vg.jpg, 7bDOOxi0KAz9GMgP3bYDhg.jpg, MymG2Zc5A9QWUE1AYuL33Q.jpg,
fp9rdA0RghLISyH9-atgyw.jpg, bBCGH3vssPXOM-gHzvL-KA.jpg, 9D4AZeD-8qPRMesOWX4OWA.jpg ...

**Candidates:**

| | |
|---|---|
| `<I1>` La Cucina Italian Eatery | `<I2>` Chuck E. Cheese |
| `<I3>` M3 Restaurant | `<I4>` Yummy's |
| `<I5>` Pizzeria Lupo | `<I6>` Brugos |
| `<I7>` Huntsman Brewing | `<I8>` Godfather's Pizza |
| `<I9>` Pizzava | `<I10>` Little Caesars |
| `<I11>` Electric Blue Elephant | `<I12>` Eclipse Pizza Co. |
| `<I13>` Knockouts Bar and Hookah Lounge | `<I14>` Firetrail Pizza |
| `<I15>` Gold 'N Silver Inn | `<I16>` J J's Pie |
| `<I17>` Smiling With Hope Pizza | `<I18>` Chicago's Pizza With A Twist Reno - S Virginia St, NV |
| `<I19>` Monaciello | `<I20>` Rick's Pizza, Beer, & More |
| `<I21>` Noble Pie Parlor - Downtown | `<I22>` Food + Drink |
| `<I23>` Korean Barbie Q | `<I24>` Pizza Baron - Reno |
| `<I25>` Semenza's Pizzeria | `<I26>` Wild Garlic Pizza & Pub |
| `<I27>` Casale's Halfway Club | `<I28>` The Pizza Collective |
| `<I29>` Pub N' Sub | `<I30>` Noble Pie Parlor - Midtown Reno |
| `<I31>` Little Waldorf Saloon | `<I32>` Mountain Mike's Pizza |
| `<I33>` Sizzle Pie | `<I34>` West Street Market |
| `<I35>` Cafe Whitney | `<I36>` Piezzetta Pizza Kitchen |
| `<I37>` Pizanos Pizza | `<I38>` Pizza Hut |
| `<I39>` Rattlesnake Club | `<I40>` zpizza |
| `<I41>` Taste of Chicago | `<I42>` Playfield 76 |
| `<I43>` Round Table Pizza | `<I44>` The Blind Onion Pizza & Pub |
| `<I45>` Opa Gourmet Pizza Cuisine | `<I46>` Paulie's Pizza |
| `<I47>` The Blind Onion Pizza & Wings | `<I48>` The Rack |
| `<I49>` Boulevard Pizza | `<I50>` MOD Pizza |
| `<I51>` Calafuria | `<I52>` Atlantis Café Alfresco |
| `<I53>` Domino's Pizza | `<I54>` Wild Garlic |
| `<I55>` Chicago's Pizza With A Twist - Reno, NV `<I56>` Peluso's Apizza | |
| `<I57>` Liberty Food & Wine Exchange | `<I58>` Opa Cafe |
| `<I59>` California Pizza Kitchen | `<I60>` Chicago Dogs! Eatery |
| `<I61>` Hidden Pizza | `<I62>` Pizza Factory |

Figure 8: A failed example of VISUALLENS on Hit@3. We list the input question, images and candidates.

# G    Limitation

While VISUALLENS demonstrates strong performance on task-agnostic multimodal recommendation, several limitations remain.

**Modular design without optimal components.** Our framework prioritizes modularity and extensibility, using a unified architecture across modules including image encoding, captioning, aspect word generation, and preference matching. However, we do not use the best-performing model for each subtask. Each component can be replaced with more advanced or domain-specific alternatives, which could potentially boost overall performance.

```
┌─────────────────────────── Output ───────────────────────────┐
│ Candidate Ranking:                                            │
│ <I40> zpizza                    <I42> Playfield 76            │
│ <I50> MOD Pizza                   <I57> Liberty Food & Wine Exchange │
│ <I33> Sizzle Pie                  <I28> The Pizza Collective  │
│ <I26> Wild Garlic Pizza & Pub     <I54> Wild Garlic          │
│ <I61> Hidden Pizza                <I36> Piezzetta Pizza Kitchen│
│ <I17> Smiling With Hope Pizza     <I5> Pizzeria Lupo          │
│ <I21> Noble Pie Parlor - Downtown <I30> Noble Pie Parlor - Midtown Reno│
│ <I12> Eclipse Pizza Co.           <I56> Peluso's Apizza       │
│ <I14> Firetrail Pizza             <I62> Pizza Factory         │
│ <I37> Pizanos Pizza               <I9> Pizzava                │
│ <I25> Semenza's Pizzeria          <I46> Paulie's Pizza        │
│ <I44> The Blind Onion Pizza & Pub <I47> The Blind Onion Pizza & Wings│
│ <I1> La Cucina Italian Eatery     <I51> Calafuria            │
│ <I19> Monaciello                  <I27> Casale's Halfway Club  │
│ <I45> Opa Gourmet Pizza Cuisine   <I58> Opa Cafe             │
│ <I59> California Pizza Kitchen    <I20> Rick's Pizza, Beer, & More│
│ <I18> Chicago's Pizza With A Twist Reno - S Virginia St,<I55> Chicago's Pizza With A Twist - Reno, NV│
│ <I41> Taste of Chicago           <I60> Chicago Dogs! Eatery  │
│ <I3> M3 Restaurant               <I22> Food + Drink          │
│ <I39> Rattlesnake Club           <I6> Brugos                 │
│ <I7> Huntsman Brewing            <I11> Electric Blue Elephant│
│ <I34> West Street Market         <I49> Boulevard Pizza       │
│ <I16> J J's Pie                  <I29> Pub N' Sub            │
│ <I48> The Rack                   <I31> Little Waldorf Saloon  │
│ <I13> Knockouts Bar and Hookah Lounge <I43> Round Table Pizza │
│ <I24> Pizza Baron - Reno         <I8> Godfather's Pizza      │
│ <I32> Mountain Mike's Pizza      <I4> Yummy's                │
│ <I35> Cafe Whitney               <I15> Gold 'N Silver Inn     │
│ <I52> Atlantis Café Alfresco     <I23> Korean Barbie Q        │
│ <I10> Little Caesars             <I38> Pizza Hut             │
│ <I53> Domino's Pizza             <I2> Chuck E. Cheese        │
└───────────────────────────────────────────────────────────────┘

┌─────────────────────────── Ground Truth ─────────────────────┐
│ <I57> Liberty Food & Wine Exchange                            │
│ <I23> Korean Barbie Q                                         │
│ <I27> Casale's Halfway Club                                   │
│ <I17> Smiling With Hope Pizza                                 │
│ <I30> Noble Pie Parlor - Midtown Reno                         │
└───────────────────────────────────────────────────────────────┘
```

Figure 9: A failed example of VISUALLENS on Hit@3. We list the output candidate ranking and ground truth. The ranking follows the same left-right order as the input candidates.

```
┌──────────────── Aspect Word Generation ────────────────┐
│ Here is a list of aspect words for a given image that the user │
│ took at a [[Category]]:                                 │
│ industrial, rust, pipes, towers, metal, structure, machinery, │
│ abandoned, overgrown, bluesky.                          │
│                                                          │
│ What are useful aspects to predict the [[Category]] the user │
│ will go after?                                          │
│                                                          │
│ Candidates of [[Category]]:                             │
│ <I1> Vulcan Park and Museum, Statue symbolizing city's  │
│ industries sits atop Red Mountain, surrounded by a park & a │
│ museum.                                                 │
│ <I2> Sloss Furnaces Visitor Center, Blast furnace plant where │
│ iron was made from 1882-1971, now an arts & education center │
│ with tours.                                             │
│ <I3> Historic Cahaba Pumping Station, ...               │
│ ...                                                     │
│                                                          │
│ Ground Truth:                                           │
│ <I2> Sloss Furnaces Visitor Center                      │
│                                                          │
│                                                          │
│ Provide the useful aspects to help predict the ground truth, │
│ separated by comma:                                     │
│ [[Answer]]                                              │
└──────────────────────────────────────────────────────────┘
```

Figure 10: The prompt template for aspect word generation.

**Limited modality and domain coverage.** Our experiments focus exclusively on static visual history (images) and do not cover more complex modalities such as video, audio, or multimodal narratives over time. These richer formats are important for building truly comprehensive, lifelong user models, as envisioned by early ideas like MEMEX [4], but remain out of scope for this work.

```
┌─────────────────────────────────────────────┐
│              Candidate Matching              │
└─────────────────────────────────────────────┘

What is the [[Category]] the user most likely to go to next?

Aspect words:
industrial, metal, structure, machinery, ...

Captions:
Image 1: A large, rusted metal structure, possibly a piece of
industrial machinery or equipment.
Image 2: Industrial landscape with tall, rusted metal
structures under a clear blue sky.
...

Candidates:
<I1> Vulcan Park and Museum, Statue symbolizing city's
industries sits atop Red Mountain, surrounded by a park & a
museum.
<I2> Sloss Furnaces Visitor Center, Blast furnace plant where
iron was made from 1882-1971, now an arts & education center
with tours.
<I3> Historic Cahaba Pumping Station, ...
...

Answer: [[Answer]]
```

Figure 11: The prompt template for candidate matching.

| Hyperparameters for training on MiniCPM-V2.5 | |
|---|---|
| Parameter Size | 8B |
| Image Resolution | $980 \times 980$ |
| Number of Image Tokens | 96 |
| Hidden Dimension Size | 4096 |
| LoRA Rank | 64 |
| LoRA $\alpha$ | 64 |
| LoRA dropout | 0.1 |
| GPU | $8 \times$ NVIDIA H100 |
| Batch Size | 8 |
| Gradient Accumulation Steps | 8 |
| Warmup Steps | 200 |
| Learning Rate | 1e-3 |

Table 10: Hyperparameters for training VISUALLENS with MiniCPM-V2.5 Backbone.

**Evaluation scope.** We restrict the recommendation task to a QA format for consistency and interpretability. While this setup allows for controlled benchmarking and comparison, it does not cover other common recommendation forms such as ranked lists, interactive dialogues, or implicit feedback modeling. Additionally, our proposed benchmarks, Google Review-V and Yelp-V, capture a subset of the full task-agnostic recommendation landscape and may not reflect the full range of real-world personalization scenarios.

Future work could explore richer modalities, more diverse recommendation formats, and further improvements to the modular components of VISUALLENS.

