# OpenReview forum: "VisualLens: Personalization through Task-Agnostic Visual History"
_NeurIPS.cc/2025/Conference — NeurIPS 2025 poster_

### Official Review · Reviewer_UTge · 2025-06-30

**Clarity:** 4
**Significance:** 3
**Originality:** 3
**Rating:** 4
**Confidence:** 3

**Summary:**

This paper proposes VisualLens, a framework that leverages MLLMs to personalize with task-agnostic visual history, i.e., user-taken photos. The method constructs a spectrum user profile using raw images, captions, and aspect words, and employs a three-step runtime pipeline: history retrieval, preference profiling, and candidate matching. The authors also introduce two new benchmarks: Google Review-V and Yelp-V, and demonstrate that VisualLens outperforms existing baselines such as UniMP and GPT-4o by 2–10% in Hit@3 and MRR. Extensive experiments validate its robustness, transferability, and effectiveness.

**Questions:**

In Figure 2, the model for aspect generation is MiniCPM-V2.5, while in 4.3.2, the author mentioned that the initial set of aspect words is generated by LLaVA-v1.6. Is it a typo?

In Line 152, the author mentioned that they number each image to ensure the images are grounded to the corresponding caption and aspect words. How these numbers are used as inputs? By editing the images?

**Ethical Concerns:**

["NO or VERY MINOR ethics concerns only"]

**Final Justification:**

I appreciate the authors' rebuttal and would like to keep my current scores as they are a proper reflection of the quality of the paper in comparison with past NeurIPS accepted papers.

**Limitations:**

Yes.

**Quality:**

3

**Strengths And Weaknesses:**

### Strengths:

- The idea or using task-agnostic visual history is interesting and novel.
- The overall framework design is clear and meet the needs for using task-agnostic visual history.
- Strong empirical results with two new, large-scale and diverse benchmarks.
- Robust and transferable performance across users, categories, and history lengths.



### Weaknesses:

- Privacy implications of using personal visual history are not discussed.
- The framework requires multiple specialized components (captioning, retrieval, gridification), each with different baseline model, which may be non-trivial to deploy in real-world settings.

---

> ### Author Rebuttal · Authors · 2025-07-30
>
> We thank the reviewer for the thoughtful feedback. We are glad that the novelty of using task-agnostic visual history for personalization, as well as the clarity and practicality of the VisualLens framework, were well appreciated. Below, we respond to the specific questions and concerns raised.
>
> **W1:** Privacy implications of using personal visual history are not discussed.
>
> **A1:** Thank you for the suggestion. We currently address privacy concerns in the **“Social Impact”** section following the Appendix. We acknowledge that real-world deployments must carefully handle user data collection, storage, and interpretation, especially when recommendations extend beyond a single domain. We will move this discussion to immediately after the main paper in the camera-ready version for greater visibility.
>
> **W2:** Modularized system is hard to deploy in real-world.
>
> **A2:** While the community increasingly favors unified systems as LLMs grow more powerful, modularized architectures still outperform in many real-world applications such as optical character recognition (OCR) and automatic speech recognition (ASR). Despite their structural complexity, modular pipelines are commonly implemented in industry, with well-established engineering practices supporting their development and maintenance.
>
> On the other hand, the modular design of VisualLens offers greater efficiency than unified models, as demonstrated below. Since the caption and aspect-word generation modules operate offline, they incur no additional cost during inference, thereby reducing runtime latency.
>
> | Model | Google-Review-V | Yelp-V |
> |------------------------|---------------------|----------------------|
> | **UniMP (3B)**   	| 5.82s            	| 4.92s            	|
> | **Visuallens (3B)**   	| 5.29s           	| 4.55s            	|
> | **Visuallens (8B)**  	 | 7.31s           	| 6.64s            	|
>
>
> **W3:** The initial set of aspect words is generated by LLaVA-v1.6. Is it a typo?
>
> **A3:** No, this is correct. The initial aspect words are generated by LLaVA-v1.6. After finalizing these aspect words, we train MiniCPM to generate them for subsequent use. We will further clarify this in the camera-ready version of the paper.
>
> **W4:** How these image numbers are used as inputs? By editing the images?
>
> **A4:** Yes, each (sub-)image is annotated with a number in the upper-left corner, which is then used as part of the input to the model. This idea is inspired by [1], which used numbering to help the model better ground to parts of an image.
>
> References:
> + [1] Yang et al., Set-of-Mark Prompting Unleashes Extraordinary Visual Grounding in GPT-4V.

---

> > ### Comment · Reviewer_UTge · 2025-08-07
> > **Thank you for the rebuttal!**
> >
> > I appreciate the authors' rebuttal and would like to keep my current scores.

---

### Official Review · Reviewer_w61j · 2025-07-02

**Clarity:** 3
**Significance:** 3
**Originality:** 2
**Rating:** 4
**Confidence:** 4

**Summary:**

This paper introduces VisualLens, a multimodal LLM-based framework for personalized recommendation using task-agnostic visual history—images from a user's daily life. It constructs a spectrum profile for each user, incorporating raw images, captions, and refined aspect words, and leverages a grid-based image aggregation for model input. The system is jointly trained for both aspect word refinement and recommendation. The authors also contribute two new benchmarks, Google Review-V and Yelp-V, and report improvements over existing SOTA models like GPT-4o and UniMP.

**Questions:**

- Would the model still work well on non-geolocated data (e.g., Pinterest photos, lifestyle vlogs)?

- How well does the method generalize to non-category-based recommendation queries?

- Is it possible to include metadata like timestamps or image locations, which are often present in photo histories?

Please also refer to the weakness part.

**Ethical Concerns:**

["NO or VERY MINOR ethics concerns only"]

**Final Justification:**

Thanks for the rebuttal. Many of my concerns are addressed. I'll keep my rating.

**Limitations:**

Yes

**Quality:**

3

**Strengths And Weaknesses:**

For strengths,
- The core idea of using casual, non-item-tied user images (e.g., photos from daily life) to personalize recommendations is both original and practically appealing.
- The method consistently outperforms state-of-the-art fine-tuned models like UniMP, and even large proprietary models like GPT-4o, on both Google Review-V and Yelp-V benchmarks—showing the benefit of dedicated adaptation.
- Google Review-V and Yelp-V are carefully curated datasets with realistic visual logs, candidate sets, and ground-truths, which will likely be valuable to the community.

For weaknesses,
- The method is only tested on geographically grounded platforms (Google Reviews, Yelp), where photos often directly relate to physical entities (e.g., restaurants, parks). It remains unclear how this method would perform in abstract or sparse-visual platforms like Amazon, where visual signals are weaker or absent for many users.
- Since the recommendation relies heavily on surface-level features like aspect words and image similarity, the method is potentially susceptible to irrelevant or adversarial content—e.g., a user uploading meme images or photos unrelated to preferences could degrade performance.
- While VisualLens combines raw images with generated captions and aspect words, both of which are textual representations, the paper does not isolate the contribution of these text components. In other words, it's unclear whether the performance gain is due to the visual content itself or the associated textual annotations. An ablation using only the captions and aspect words—excluding the image—would help clarify whether visual signals contribute meaningfully beyond their textual projections.

---

> ### Author Rebuttal · Authors · 2025-07-30
>
> We thank the reviewer for the thoughtful feedback. We appreciate the recognition of our novelty, performance improvement against SOTA models, including GPT-4o and UniMP, as well as the value of our newly introduced Google Review-V and Yelp-V benchmarks. Below, we provide clarifications and address the questions and suggestions raised.
>
> **W1:** Would the model still work well on non-geolocated data?
>
> **A1:** We agree task-agnostic personalized QA on non-geolocated data, such as lifestyle vlogs, is an interesting future direction.Notably, Google Review-V already contains some lifestyle-oriented images (e.g., a cat in the garden), suggesting that VisualLens can generalize to such settings, provided that a sufficient number of relevant images are available (typically around 50, as shown in Figure 3).
>
> However, due to personal data privacy concerns, creating a public benchmark for this setting remains challenging. We have discussed this in our Limitation (L615).
>
> **W2:** The method is potentially susceptible to irrelevant or adversarial content.
>
> **A2:** VisualLens begins with an image retrieval step, which helps **filter out irrelevant content before personalization**, reducing the risk of noisy or adversarial inputs. As shown in our ablation studies in Table 4, this retrieval step plays a critical role in maintaining performance. Furthermore, Figure 3 illustrates that performance degrades as the proportion of relevant images decreases. However, **MRR stabilizes once the number of relevant images reaches approximately 50**, demonstrating a degree of robustness under typical usage conditions.
>
> **W3:** How well does the method generalize to non-category-based recommendation queries?
>
> **A3:** Most recommendation queries implicitly or explicitly involve a category, ranging from coarse-grained ones like "restaurant" to finer-grained ones like "Italian restaurant". In such cases, the category mainly defines the candidate set, and **VisualLens remains applicable regardless of granularity.** Our evaluation benchmarks include broad categories such as "area", which resemble open-ended queries like "give me an idea on where to go?" (see Figure 4 for the full category list). That said, extremely general prompts like "recommend something" fall outside the scope of this paper.
>
> **W4:** An ablation using only the captions and aspect words—excluding the image—would help clarify whether visual signals contribute meaningfully beyond their textual projections.
>
> **A4:** Thank you for the suggestion. We address this in **Table 3, Row 4**. Since Visuallens without image input cannot generate aspect words for joint fine-tuning, we instead evaluate a text-only variant of Visuallens-8B. In this variant, the same backbone LLM (Llama-3.1-8B-Instruct) is fine-tuned using only captions and aspect words. Compared to the full Visuallens-8B model, this leads to an MRR drop of **2.5 on Google-Review-V and 5.5 on Yelp-V**, highlighting the significant contribution of visual signals.
>
> **W5:** Is it possible to include metadata like timestamps or image locations, which are often present in photo histories?
>
> **A5:** Yes, incorporating metadata such as timestamps or location is possible in principle. However, due to the limited context length of our VLM backbone (MiniCPM), we prioritized optimizing image representations. Including metadata is a promising direction for future work.

---

> > ### Comment · Reviewer_w61j · 2025-08-05
> >
> > Thanks for the detailed response, which has addressed my concerns. I’ll maintain my rating.

---

### Official Review · Reviewer_bdom · 2025-07-02

**Clarity:** 3
**Significance:** 4
**Originality:** 3
**Rating:** 5
**Confidence:** 3

**Summary:**

The authors present a system of multimodal language models for content recommendation. They intuitively propose a reasonable motivation and an application plan for multimodal large models in recommendation systems, aiming to improve the accuracy of the recommendation system by using users’ visual history as reference information. To this end, the authors introduce VisualLens, which functions like a data annotation pipeline. By leveraging the powerful understanding capabilities of multimodal large models, it re-encodes and compresses historical record information to assist in ranking the recommended content. The authors also propose a corresponding benchmark to validate the effectiveness of the method.

**Questions:**

- There are too many `Avg. # of images` in Table 2, is the model able to accept so much tokens?
- It their any comparison to traditional recommandation system?

**Ethical Concerns:**

["NO or VERY MINOR ethics concerns only"]

**Final Justification:**

I keep my rating and confidence. The authors resolve my confusion about model sizes, token number and comparison method. I consider this paper advance the MLLM application on recommandation system.

**Limitations:**

- It seems that the content is still image-level, while i consider the recommandation is more important for video recommandation.
- Maybe a unified model can be implemented by getting embeddings from VLM, (like using VLM2Vec(ICLR2025)?)
Because the model you used whose ViT is not pre-trained with contrastive loss (for example: NTP loss), the matching may be misaligned.

**Quality:**

3

**Strengths And Weaknesses:**

Strengths:

- I agree with the motivation of this paper, which is to build an intelligent agent to understand user interests and habits for making recommendations. I think this promotes the application of multimodal large models in the next generation of recommendation systems. I believe this could help change the current state of the field, where contrastive learning is primarily used to apply multimodal image-text models in real-world applications.
- Good performance and lower cost: their implementation achieved better performance while using a small MiniCPM.
- Detailed description about the pipeline.

Weaknesses:
- It seems that the authors do not decide to open-source the data and implementation?
- The 8B model is still big, a large amount of daily active user data on many platforms is difficult to process and infer quickly.

---

> ### Author Rebuttal · Authors · 2025-07-30
>
> We thank the reviewer for the thoughtful and encouraging feedback. We are glad that the motivation behind VisualLens is well received. We appreciate the recognition of our system's practical relevance, the performance improvement, and the detailed design of our pipeline. Below, we respond to the specific questions and concerns raised.
>
> **W1:** Open-source.
>
> **A1:** We plan on open-sourcing the data and implementation (pending legal approval).
>
> **W2:** The 8B model is still big and hard to process data quickly.
>
> **A2:** We agree that 8B models can be resource-intensive. However, it already applies in chatbot use cases, which has tolerance to latency from large models.
> In addition, VisualLens also performs well with a 3B model, and still **outperforms UniMP of the same size in both accuracy and latency** (results on a single NVIDIA L40S GPU below).
>
> | Model | Google-Review-V | Yelp-V |
> |------------------------|---------------------|----------------------|
> | **UniMP (3B)**   	| 5.82s            	| 4.92s            	|
> | **Visuallens (3B)**   	| 5.29s           	| 4.55s            	|
> | **Visuallens (8B)**  	 | 7.31s           	| 6.64s            	|
>
> As the first study on task-agnostic visual recommendation, our goal is to establish a strong foundation. We anticipate future work exploring smaller models, efficient architectures, and even mobile-friendly deployments.
>
>
> **W3:** Avg. # of user history images is large, how to handle so many tokens?
>
> **A3:** To address VLM context length limitations, we apply image filtering and gridification on user history images, ensuring each example is encoded as a single gridified image with no more than 64 sub-images. This allows us to remain within context limits while retaining informative visual content.
>
> **W4:** Comparison to traditional recommandation system?
>
> **A4:** In Table 3, we compare VisualLens against one of the strongest traditional multimodal recommendation systems, UniMP. We also benchmark against various LLM-based recommendation methods, including both fine-tuned and prompting-based approaches.
>
> **W5:** Mostly image-level, no video-level.
>
> **A5:** We appreciate the reviewer pointing out this important direction for future research. While our current focus is on image-based recommendation, which has direct applications in e-commerce, travel planning, and social media, we agree that extending to video is a valuable next step. We have discussed  this in the Limitation section.
>
> We consider our work on static images to be a foundational step. Many state-of-the-art video analysis techniques rely on robust image-level understanding as a core component. One future study can be applying VisualLens to video keyframes and aggregating features over time.
>
> **W6:** Getting embeddings from VLM to implement a unified model?
>
> **A6:** Thank you for your suggestion! We find visual image representations alone are not sufficient for task-agnostic recommendation. This motivated our modular approach using joint image representations to capture more nuanced signals. We agree exploring the possibility of a unified system through better embedding is another promising direction. We will add this to the Limitations section and include related works such as VLM2Vec in our discussion.

---

> > ### Comment · Reviewer_bdom · 2025-08-01
> >
> > Thanks for the authors' response. I currently have no more question.

---

> > > ### Author Response · Authors · 2025-08-04
> > >
> > > Thank you for your time and for engaging with our work. We're glad we could address your questions. Please feel free to reach out if anything else comes up.

---

### Official Review · Reviewer_bz3j · 2025-07-20

**Clarity:** 3
**Significance:** 2
**Originality:** 2
**Rating:** 4
**Confidence:** 4

**Summary:**

In this paper, the authors proposed VisualLens, a personalized‑recommendation framework that leverages a user’s task‑agnostic visual history. The method employs multimodal large language models to distill each image into a compact “spectrum” representation, forming a rich user profile. To benchmark the approach, the authors release two large‑scale datasets—Google Review‑V and Yelp‑V—and show that VisualLens delivers robust, state‑of‑the‑art performance across both.

**Questions:**

1. Could you specify which components of your pipeline are fundamentally new beyond CLIP4Clip (2023) and UniMP (2024), and provide an ablation showing the exact contribution (e.g., Hit@k delta) of each?
2. Have you run any human double‐annotation or LLM‑as‑Judge audits on a stratified sample of Google Review‑V and Yelp‑V to estimate label noise or inter‑annotator agreement?

**Ethical Concerns:**

["NO or VERY MINOR ethics concerns only"]

**Final Justification:**

Thanks for the thorough experiments. Many of my concerns are addressed.

**Limitations:**

yes

**Paper Formatting Concerns:**

NAN

**Quality:**

3

**Strengths And Weaknesses:**

Pros:

1. The topic is both intriguing and holds potential for significant developments.
2. The paper is well-organized and reader-friendly, making it easy to understand.
3. The experiment results seem promising

Cons:

1. The pipeline (CLIP retrieval → image‑grid prompt → LoRA‑tuned MLLM) replicates CLIP4Clip/UniMP, and the only new “aspect‑word” tweak lifts Hit@3 by < 2 pp, so novelty is incremental.
2. Google Review‑V and Yelp‑V use raw user review tags without any human or LLM audit
3. Results cover only oracle re‑ranking accuracy; the paper omits retrieval‑recall, latency, and etc

---

> ### Author Rebuttal · Authors · 2025-07-30
>
> We thank the reviewers for recognizing the potential impact of VisualLens in advancing personalized recommendation through task-agnostic visual histories. We appreciate the constructive feedback and, below, address the raised concerns and offer clarifications.
>
> **W1:** Which components of your pipeline are fundamentally new beyond CLIP4Clip (2023) and UniMP (2024), and provide an ablation?
>
> **A1:** Thank you for the thoughtful question. First, we would like to clarify that CLIP4Clip (2021) is a video-text retrieval model, while UniMP (2024) proposes a unified multimodal recommendation system trained via multi-task learning. In contrast, VisualLens tackles a fundamentally different problem: **task-agnostic visual recommendation**, which leverages a user’s general visual history (e.g., everyday photos) instead of task-specific interaction data. This setting presents new challenges, as conventional **weak image-text representations (e.g., used in UniMP)** fail to generalize effectively.
>
> VisualLens introduces several key innovations beyond CLIP4Clip and UniMP:
>
> **1. New problem setting:** We formulate the task of task-agnostic visual recommendation, where the user history is broad and not tied to a specific domain or task.
>
> **2. Spectrum image representation:** We enrich raw visual features with captions and aspect words, creating a spectrum of representations that capture varying levels of abstraction:
> + Visual (dense but noisy),
> + Captions (moderately abstract),
> + Aspect words (sparse but precise).
>
> Our ablations in Table 4 demonstrate that each modality contributes meaningfully to performance.
>
> **3. Gridification of images:** To fit within the limited context window of multimodal LLMs, we propose a gridification strategy to encode multiple images efficiently. This is not addressed in CLIP4Clip or UniMP.
>
> **4. Joint fine-tuning:** We jointly train the aspect word generation and the preference prediction modules, improving both performance and efficiency, unlike prior works that rely on disjoint stages or fixed encoders.
>
> **Empirical support:**
>
> We compare **VisualLens directly with UniMP** using the same retrieval setup in **Table 3, Row 3**. VisualLens-3B consistently outperforms UniMP across both benchmarks:
> + MRR: +3.0% (Google Review-V), +5.0% (Yelp-V)
> + Hit@3: +2.2% (Google Review-V), +10.3% (Yelp-V)
>
> Further ablation studies in Table 4 show that removing captions and aspect words causes VisualLens to underperform UniMP. Notably, removing just the aspect words leads to a **statistically significant drop**:
> + Google Review-V: -1.3% (Hit@3, MRR)
> + Yelp-V: -3.6% (Hit@3), -2.5% (MRR), with p < 0.05
>
> These results highlight the importance of each module in our system and the necessity of robust joint representations for task-agnostic recommendation.
>
> Lastly, we are not aware of a 2023 version of CLIP4Clip. If the reviewer is referring to a newer paper, we would appreciate a reference and are happy to address it. Regardless, we will cite CLIP4Clip (2021) in our related work to properly acknowledge its relevance to retrieval-based multimodal systems.
>
>
> **W2:** Have you run any human double-annotation or LLM-as-Judge audits on a stratified sample of Google Review-V and Yelp-V to estimate label noise or inter-annotator agreement?
>
> **A2:** We did not conduct additional human annotations. However, we note that both Google Local Data and Yelp are widely used in the recommendation literature (e.g., [1], [2], [3]). Our work does not introduce new labels but restructures existing ones to support the task-agnostic visual recommendation benchmark.
>
> Besides, to reduce potential noise, we filter out businesses with fewer than 10 reviews, ensuring that each business label is supported by multiple human annotations. Additional details are provided in Appendix A.
>
> References:
> + [1] Li et. al., UCTopic: Unsupervised contrastive learning for phrase representations and topic mining. ACL 2022.
> + [2] Yan et al., Personalized showcases: Generating multi-modal explanations for recommendation. SIGIR 2023.
> + [3] Mezentsev et al., Scalable Cross-Entropy Loss for Sequential Recommendations with Large Item Catalogs. RecSys 2024.
>
> **W3:** Results cover only oracle re‑ranking accuracy; the paper omits retrieval‑recall, latency.
>
> **A3:** For retrieval‑recall, ground truth for retrieval is not available, so we cannot report retrieval recall directly. In addition, recommendation tasks are fairly robust against small retrieval quality differences (as reported in Table 6). Table 4 ablation results suggest the retrieval stage is both necessary and effective, removing it leads to a 6.6% drop in MRR for Google Review-V and 6.1% drop for Yelp-V.
>
> Regarding latency, following reviewer’s suggestion, we measured inference wall-clock time per example on a single NVIDIA L40S GPU. The results demonstrate that the joint image representation learned by VisualLens is more **efficient and effective** than the interleaved multimodal representation used in UniMP, reducing latency by ~0.5 sec.
>
> | Model | Google-Review-V | Yelp-V |
> |------------------------|---------------------|----------------------|
> | **UniMP (3B)**   	| 5.82s            	| 4.92s            	|
> | **Visuallens (3B)**   	| 5.29s           	| 4.55s            	|
> | **Visuallens (8B)**  	 | 7.31s           	| 6.64s            	|
>
> We will include these new latency results in the camera-ready version of the paper.

---

> > ### Comment · Reviewer_bz3j · 2025-08-04
> > **response to rebuttal**
> >
> > Thanks for the thorough experiments. Many of my concerns are addressed; I’ll increase my score.

---

> > > ### Author Response · Authors · 2025-08-04
> > >
> > > Thank you for your updated feedback and for carefully reviewing our work. We're glad the additional experiments helped address your concerns, and we sincerely appreciate your willingness to revise your score.

---

### Comment · Area_Chair_hyrA · 2025-08-05
**Please provide your feedback on the authors' rebuttal.**

Dear reviewers,

For those who have not responded yet, please take a look at the authors’ rebuttal and update your final scores.

Best wishes,

AC

---

### Decision · Program_Chairs · 2025-09-17

**Decision:**

Accept (poster)

**Comment:**

The reviews (5,4,4,4) for this paper have been collected and discussed. There is a general consensus among the reviewers that the paper, in its current form, is suitable for publication. After careful consideration of the feedback and the paper itself, the recommendation is to accept this submission.